# Rare heterozygous missense variants in *VSX2* are associated with retinal detachment

Daniel C. Brock[1,2�], Justin S. Dhindsa[1,2�], Yifan Chen[1,2,3,4], Vida Ravanmehr[3], Jonathan Mitchell[5], Fengyuan Hu[5], Xiaoyin Li[6], Likhita Nandigam[3], Quanli Wang[6], Kevin Wu[1], Jessica C. Butts[7], Hardeep S. Dhindsa[8], Benjamin J. Frankfort[1,9,10], Nicholas M. Tran[1], Slavé Petrovski[5,11]*, Ryan S. Dhindsa[1,3,4]*

1 Department of Molecular and Human Genetics, Baylor College of Medicine, Houston, Texas, United States of America, 2 Medical Scientist Training Program, Baylor College of Medicine, Houston, Texas, United States of America, 3 Department of Pathology and Immunology, Baylor College of Medicine, Houston, Texas, United States of America, 4 Jan and Dan Duncan Neurological Research Institute, Texas Children's Hospital, Houston, Texas, United States of America, 5 Centre for Genomics Research, Discovery Sciences, R&D, AstraZeneca, Cambridge, United Kingdom, 6 Centre for Genomics Research, Discovery Sciences, R&D, AstraZeneca, Waltham, Massachusetts, United States of America, 7 Department of Bioengineering, Neuroengineering Initiative, Rice University, Houston, Texas, United States of America, 8 HD Retina Eye Center, Reno, Nevada, United States of America, 9 Department of Ophthalmology, Baylor College of Medicine, Houston, Texas, United States of America, 10 Department of Neuroscience, Baylor College of Medicine, Houston, Texas, United States of America, 11 Department of Medicine, Austin Health, University of Melbourne, Melbourne, Australia

☦ These authors contributed equally to this work.
* ryan.dhindsa@bcm.edu (RSD); slav.petrovski@astrazeneca.com (SP)

## Abstract

Retinal detachment (RD) is a sight-threatening emergency requiring urgent intervention to prevent permanent vision loss. While both environmental and genetic risk factors contribute to RD, its complete genetic architecture remains unknown. Here, we performed the largest whole genome sequencing-based case-control study in RD to date, including data from 7,276 RD cases and 236,741 controls in the UK Biobank. Through variant- and gene-level association analyses, we identified *VSX2* as a genetic determinant of RD risk while confirming established associations including *FAT3*, *RDH5*, and *COL2A1*. Gene-level collapsing analysis revealed that rare heterozygous missense variants in *VSX2* confer a 2.8-fold increased risk of RD (p = 2.4x10$^{-10}$; odds ratio (OR) = 2.8; 95% confidence interval (CI): [2.1, 3.7]). One missense variant in this gene, p.Glu218Asp, demonstrated a particularly strong effect size (p = 9.3x10$^{-10}$; OR = 5.9; 95% CI: [3.7, 9.4]). Replication analyses in two additional cohorts, totaling 1,331 cases and 52,355 controls strengthened both the gene- and variant-level associations even further (p = 1.4x10$^{-10}$ and 1.1x10$^{-11}$, respectively). Other contributory heterozygous variants included previously reported pathogenic homozygous variants for anophthalmia and microphthalmia. These findings thus reveal a previously unknown gene dosage curve for *VSX2*, where homozygous mutations cause severe developmental eye disorders and heterozygous mutations cause adult-onset retinal detachment. Extending this observation, we found a significant enrichment for other

**Data availability statement:** Code availability: Association tests were performed using a custom framework, PEACOK, available on GitHub (https://github.com/astrazeneca-cgr-publications/PEACOK/). Data availability: The ExWAS and gene-level collapsing analysis generated in this study are derived from whole genome sequencing from the UKB, which is publicly available for registered users. The genome data is available in the UKB Showcase Portal: https://biobank.ndph.ox.ac.uk/showcase/. All summary statistics from the association tests are publicly available through the AstraZeneca Centre of Genomics Research PheWAS Portal (http://azphewas.com/). Raw and normalized count matrixes for snRNA-seq data can also be downloaded using the CELLxGENE collection52. The dataset for developing human retina from Zuo et al59 can be downloaded from CELLxGENE (https://cellxgene.cziscience.com/collections/5900dda8-2dc3-4770-b604-084eac-1c2c82) or GEO with the accession code GSE268630 (https://www.ncbi.nlm.nih.gov/geo/query/acc.cgi?acc=GSE268630).

**Funding:** This study was carried out with support of the National Institute of Health (DP5OD036131 to R.S.D; R01EY036111 to N.M.T; R01EY035646 to B.J.F), the Norn Group, Hevolution Foundation, and Rosenrankz Foundation (Longevity Impetus Grant to R.S.D), Texas Children's Hospital Research Vision Scholar Program (to R.S.D), the Robert and Janice McNair Foundation (to D.C.B), The Cullen Foundation (to J.S.D), and Rice University start-up funds (to J.C.B). The funders had no role in study design, data collection and analysis, decision to publish, or preparation of the manuscript.

**Competing interests:** I have read the journal's policy and the authors of this manuscript have the following competing interests: J.M., F.H., X.L., Q.W, and S.P. are current employees and/or shareholders of AstraZeneca. R.S.D. has received consulting fees from AstraZeneca.

known recessive Mendelian eye disease genes among nominally significant ($p < 0.05$) genes associated with RD in the collapsing analysis. This work provides a compelling example of how heterozygous variants in recessive disease genes can be associated with less severe clinical phenotypes.

## Author summary

Retinal detachment (RD) is a medical emergency that can lead to permanent blindness if not treated promptly. Although both environmental and genetic factors contribute to RD risk, much of its genetic basis has remained unclear. In this study, we analyzed whole genome sequencing data from more than 240,000 individuals in the UKBiobank, representing the largest genetic study of RD to date. We discovered that rare variants in *VSX2*, a gene essential for retinal development, substantially increase RD risk in adults, independent of known risk factors such as myopia or cataract. We further confirmed this association in two independent cohorts: All of Us and 100kGP. While homozygous *VSX2* variants are known to cause severe childhood eye conditions like microphthalmia or anophthalmia, our findings demonstrate that carrying a single altered copy of *VSX2* increases susceptibility to RD later in life, expanding the phenotypic spectrum of this gene. We also found that heterozygous variants in other genes linked to recessive eye diseases may similarly increase RD risk. Collectively, these findings reveal connections between rare developmental disorders and common adult disease, illustrating how large-scale genomic analysis can uncover hidden contributors to vision loss and inform future strategies for early detection.

## Introduction

Retinal detachment (RD) is a vision-threatening condition which can require urgent surgical intervention to prevent permanent blindness [1]. RD occurs when adhesion forces between the neurosensory retina and retinal pigment epithelium (RPE) are disrupted, leading to the accumulation of subretinal fluid and subsequent detachment of the two layers. RD can be classified into distinct types based on the underlying pathophysiology. The most common form is rhegmatogenous RD, where retinal tears enable fluid accumulation in the subretinal space. Serous/exudative RD is a rarer class of RD and is characterized by subretinal fluid accumulation without retinal tears due to inflammatory, neoplastic, or vascular pathologies [2,3]. Tractional RD is a third form of RD that can occur when vitreous forces elevate the retina from the RPE, often seen in patients with diabetes, previous trauma, or following surgical interventions [4]. Known risk factors for rhegmatogenous RD include high myopia, advanced age, male sex, ocular trauma, intraocular surgery, and genetic predisposition [5,6]. While prompt surgical intervention achieves successful reattachment in over 90% of cases, genetically predisposed individuals present unique challenges, including

earlier onset, larger tears, frequent recurrence, and bilateral involvement [7–9]. Understanding these genetic risk factors is therefore critical for both prevention and development of targeted therapies.

To date, our understanding of RD genetics has emerged from two main sources: studies of rare Mendelian syndromes that feature RD among their manifestations and genome-wide association studies (GWAS) of more common forms of RD. Mendelian diseases have implicated 29 genes [10], many of which encode proteins involved in vitreous connective tissue formation and homeostasis [11]. For example, mutations in collagen genes including *COL2A1*, *COL11A1*, and *COL11A2*, cause Stickler syndrome, characterized by rhegmatogenous RD alongside systemic connective tissue manifestations [12,13]. However, monogenic causes represent only a small minority of rhegmatogenous RD, likely accounting for ~1% of all cases [14]. Most RD cases arise from a combination of genetic variants and environmental or anatomical risk factors, like myopia [10]. Correlation analyses from prior RD GWAS support a shared genetic architecture between RD and its major epidemiologic risk factors, including high myopia and cataract operation [15]. Nonetheless, first-degree relatives of RD patients have approximately twice the lifetime risk of rhegmatogenous RD, independent of myopia [16]. Prior GWAS have identified numerous RD-associated loci, most of which are highly expressed in the neural retina or retinal pigmented epithelium [15,17]. Furthermore, many of the loci are broadly involved in extracellular matrix, cytoskeletal, and cell-adhesion pathways [10]. However, common variants only explain between 23–27% of RD heritability [14,15], suggesting the involvement of rare variants.

Rare variants typically confer larger effect sizes and can provide clearer mechanistic and therapeutic insights [18]. The advent of population-scale biobanks with linked exome/genome sequencing data and health records offers unprecedented opportunities to investigate rare variant associations with many understudied complex traits. Using these data, we and others have demonstrated a clear role of rare variants in thousands of complex traits, many of which occur in genes previously linked to rarer, Mendelian forms of disease [19–21]. These datasets have also begun to provide evidence of a long-suspected paradigm that heterozygous carriers of many recessive disease variants may exhibit attenuated phenotypes, distinct from the more severe manifestations seen in homozygous individuals [22,23]. While this has been documented in a handful of disorders, including retinal dystrophies and cataracts, the potential contribution of this mechanism to RD risk remains unexplored [22,23].

Here, we leveraged whole genome sequencing (WGS) data with paired clinical data from nearly 500,000 UK Biobank (UKB) participants to expand our understanding of the genetic architecture of RD. Through variant- and gene-level analyses, we uncovered a novel association driven by rare missense variants in *VSX2* (formerly *CHX10*), which encodes a conserved transcription factor that is required for neural retinal development and maintenance [24–27]. Homozygous missense and loss-of-function mutations in *VSX2* cause congenital microphthalmia/anophthalmia [28–38], coloboma, cataracts, optic nerve hypoplasia, and non-progressive retinal scotomas [39]. To our knowledge, this is the first report that heterozygous mutations in *VSX2* confer a substantial increased risk of RD. Extending this observation, we show that other top-ranking genes from the RD collapsing analysis are significantly enriched for recessive eye disease genes. These results expand the phenotypic spectrum of *VSX2*-related disorders, highlight the contribution of rare variants to RD, and expand on the overlap between Mendelian and complex genetic architectures.

## Results

### Study design and demographics

We processed genome sequence data from 490,560 multi-ancestry UKB participants through our previously described cloud-based pipeline [19]. Quality control was performed by removing samples with low sequencing quality and from closely related individuals (Methods). We identified individuals with retinal detachment (RD) by aggregating self-reported data, International Classification of Diseases (ICD10) codes from electronic health record billing codes, and death registry data (Methods). For our primary analysis, we aggregated ICD-10 code H33 and all of its sub-codes. In total, we identified 7,593 RD cases, including 7,276 of European ancestry (EUR), 128 of South Asian ancestry (SAS), 92 of African ancestry

(AFR), 27 of East Asian ancestry (EAS), 61 of Ashkenazi Jewish ancestry (ASJ), and 47 of admixed American ancestry (AMR) (**Fig 1**). Among the EUR cohort, the median age of RD diagnosis was 60 (Interquartile range [IQR]: 53–67); 4,087 (56%) were male and 3,187 (44%) were female. To define controls, we excluded individuals who had any diagnosis under ICD-10 Chapter VII (diseases of the eye and adnexa). We sex-matched our controls, resulting in 251,496 controls, including 236,741 EUR, 4,122 SAS, 4,995 AFR, 1,437 EAS, 1,462 ASJ, and 2,739 AMR samples (**Fig 1**) [19].

**Variant-level ExWAS**

We first performed a variant-level exome-wide association study (ExWAS) in the UKB EUR cohort to identify protein-coding variants associated with RD risk. Unlike GWAS, which are limited to common variants and rarer imputed variants, this approach allows us to test for associations across the allele frequency spectrum (minor allele count > 5) [19]. Genomic inflation was well-controlled across the tested additive ($\lambda_{GC}$: 1.02), dominant ($\lambda_{GC}$: 1.02), and recessive ($\lambda_{GC}$: 0.94) genetic models (**S1 Fig**).

There were four significant ($p < 1\times10^{-8}$) associations across three genes: *FAT3*, *VSX2*, and *RDH5* (**Table 1 and Fig 2A**). *FAT3* and *RDH5* have previously been associated with RD and myopia, respectively, via GWAS [15]. Several other well-established GWAS loci ranked among the top hits in our exome-wide association study (ExWAS), although they fell just below the study-wide significance threshold, including *TYR*, *BMP3*, *PLCE1*, and others (**S1 Table**). These positive control associations also validate the robustness of the phenotype definitions.

To the best of our knowledge, *VSX2* variants have not been previously associated with RD. This association emerged from a rare missense variant (p.Glu218Asp; 14−74259676-G-T) under the dominant model ($p = 9.3\times10^{-10}$; OR [95% CI] = 5.9 [3.7-9.4]). Its minor allele frequency (MAF) in population controls is 0.02% (n = 116/236,659) compared to the 0.14% observed in cases (n = 21/7,274). Similar to our UK Biobank European population controls, its MAF is 0.02% in the independent non-Finnish European subset of gnomAD v2.1 (n = 64,603) [40]. Thus, this variant was likely too rare to be detected in prior GWAS studies of RD. Notably, this variant confers nearly 6-fold increased odds of developing RD, an effect size demonstrably larger than the associations driven by more common variants (**Table 1**). Interestingly, recessive loss-of-function and missense variants in *VSX2* have previously been associated with microphthalmia/anophthalmia that presents at birth [28,29,31–33].

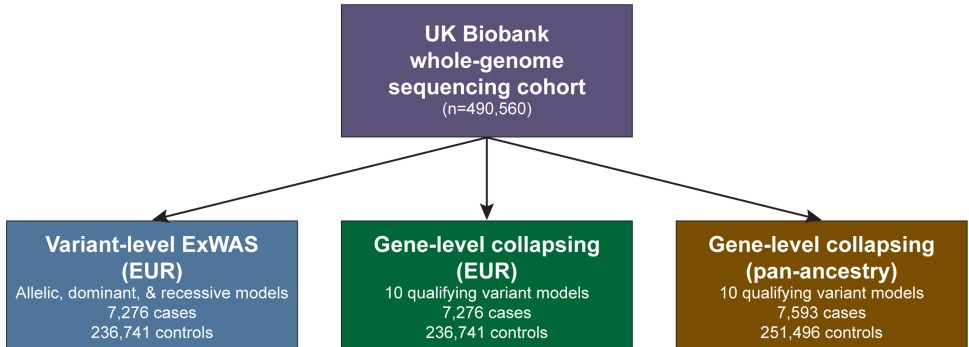

**Fig 1. Genetic association study design.** The discovery cohort comprised whole genome sequencing data from UK Biobank participants. Variant-level ExWAS and gene-level collapsing analyses were conducted in the European (EUR) cohort. Gene-level collapsing analysis was repeated in a pan-ancestry population, consisting of individuals from European, African, East Asian, and South Asian ancestries.

## Gene-level collapsing analysis

We next applied our gene-level collapsing analysis framework, which tests rare variants in aggregate and bolsters power for rare variant-driven associations [19,20,41]. Briefly, we identify rare variants that meet a predefined set of criteria (that is, 'qualifying variants' or 'QVs') in each gene and test for their aggregate effect. We tested a total of 18,930 genes under 10 different QV models to evaluate a range of possible rare-variant genetic architectures (S2 Table). We performed one analysis on individuals filtered for European ancestry (~90% of the UKB) and another pan-ancestry analysis that included individuals from all represented ancestries in the UKB to ensure we maximize the value of the diversity in this resource (Methods). As with our ExWAS results, we observed no evidence of genomic inflation in our test statistics (S2 Fig; median $\lambda_{GC} = 1.02$; range: 0.99, 1.03).

COL2A1 and VSX2 were significantly associated with RD in the UKB European collapsing analysis (Tables 1 and S3 and Fig 2B). COL2A1 was most significantly associated with RD under the ptv model, which exclusively assesses the contribution of rare (MAF ≤ 0.1%) protein-truncating variants (PTVs) (p = 3.45x10$^{-14}$; OR [95% CI] = 162.9 [35.7-743.7]). COL2A1 is a well-established gene for Stickler syndrome, a rare hereditary connective tissue disorder characterized by impaired production of collagen II, IX, and XI [10]. Approximately 70% of patients with COL2A1-associated Stickler syndrome develop retinal detachment [12], making COL2A1 an appropriate positive control. Moreover, the OR of 162.9 exemplifies the profound effects that COL2A1 haploinsufficiency has on eye health. The significant association between COL2A1 with RD in the collapsing analysis, but not in ExWAS, also underscores the increased statistical power gained from aggregating the effects from a series of large-effect and (ultra-) rare variants across a gene (Fig 2C).

VSX2 was also significantly associated with RD in the collapsing analysis under the 'flexnonsynmtr' QV model (p = 2.4x10$^{-10}$; OR [95% CI] = 2.8 [2.1-3.7]). This model incorporates two types of rare variants (MAF ≤ 0.1%): the combination of protein-truncating variants and missense variants located in missense-intolerant genetic sub-regions of VSX2, as determined by the missense tolerance ratio (MTR) [42]. The c.654G>T (p.Glu218Asp) variant in the ExWAS contributed as a QV in this model. To investigate whether other VSX2 variants contributed additional signal beyond the c.654G>T variant, we re-ran the collapsing analysis excluding carriers of this variant, which corresponded to the removal of 21 out of 54 qualifying cases and 116 out of 634 qualifying controls. The collapsing analysis signal remained significant (p = 2.1x10$^{-4}$, OR [95% CI] = 2.1 [1.4-3.0]), indicating the contribution of additional rare deleterious variants in VSX2 are associated with RD (Fig 3A).

**Table 1. Significant associations from UKB ExWAS and gene-level collapsing analysis.**

| Analysis (model) | Gene | Genotype (AA change) | *P*-value | Odds Ratio [95% CI] | Case freq. | Control freq. |
|---|---|---|---|---|---|---|
| ExWAS (allelic) | FAT3 | 11-92840745-G-T (p.Val3518Leu) | 1.0x10$^{-24}$ | 0.8 [0.8-0.9] | 0.36 | 0.40 |
| **ExWAS (dominant)** | **VSX2** | **14-74259676-G-T (p.Glu218Asp)** | **9.3x10$^{-10}$** | **5.9 [3.7-9.4]** | **0.0014** | **2.5x10$^{-4}$** |
| ExWAS (allelic) | FAT3 | 11-92857282-A-G (p.Ser3812Gly) | 1.9x10$^{-9}$ | 0.9 [0.85-0.92] | 0.23 | 0.21 |
| ExWAS (allelic) | RDH5 | 12-55721801-C-T (p.Ile141=) | 8.5x10$^{-9}$ | 0.9 [0.86-0.93] | 0.22 | 0.24 |
| Collapsing (*ptv*) | COL2A1 | N/A | 3.5x10$^{-14}$ | 162.9 [35.7-743.7] | 0.0014 | 8.4x10$^{-6}$ |
| **Collapsing (*flexnonsynmtr*)** | **VSX2** | **N/A** | **2.4x10$^{-10}$** | **2.8 [2.1-3.7]** | **0.0074** | **0.0027** |

Associations that reached p < 1x10$^{-8}$ in the ExWAS or collapsing analysis in the UK Biobank. Full results are available as S1 and S3 Tables. Collapsing analysis models are defined in S2 Table. AA = amino acid; CI = confidence interval; freq. = frequency.

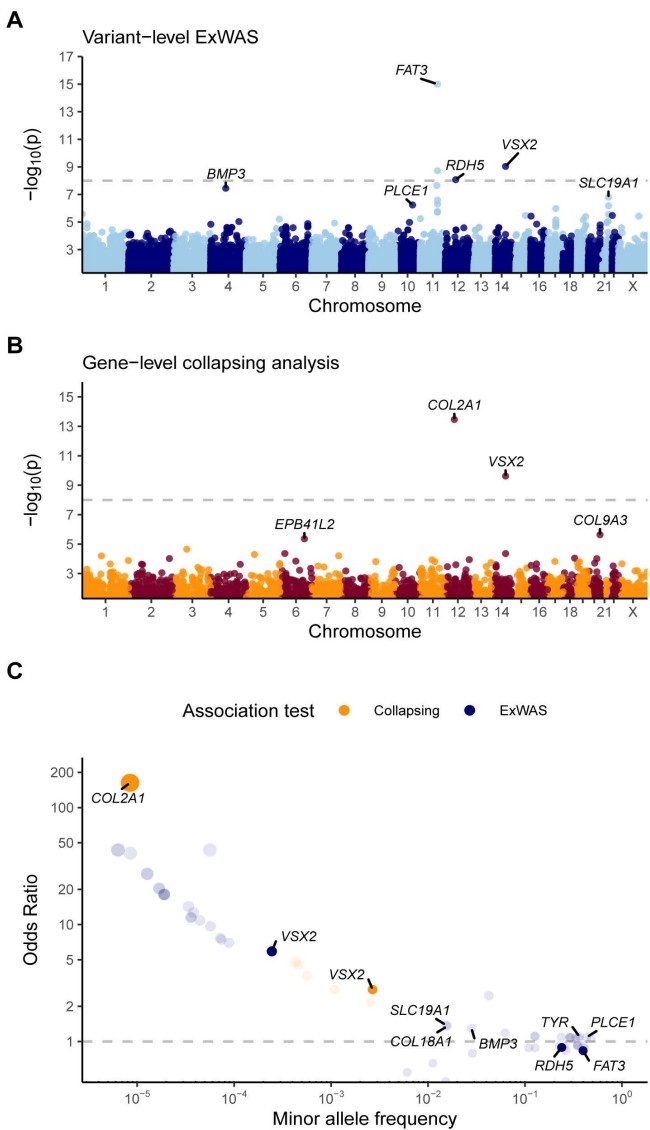

**Fig 2. Variant- and gene-level associations with RD.** Manhattan plot for **(A)** variant-level ExWAS analysis and **(B)** gene-level collapsing analysis. Horizontal dashed lines represent the p-value threshold for determining statistical significance (p ≤ 1x10⁻⁸). For variants/genes with multiple models associated with RD, the model with the most significant p-value was displayed. The p-values were capped at 1x10⁻¹⁵ for plotting clarity. **(C)** Effect sizes for RD-associated genes, plotting minor allele frequency versus odds ratio. Transparent points represent associations with p < 5x10⁻⁵ and solid points represent significant associations (i.e., p < 1x10⁻⁸). Genes are colored according to the respective analysis in which they were found to be statistically significant. Note: *VSX2* was found to be statistically significant in both ExWAS and collapsing analyses.

Finally, we performed a pan-ancestry collapsing analysis, which recapitulated the top-ranking hits from the European ancestry group: *COL2A1* and *VSX2* (**S4 Table**). While no additional genes achieved statistical significance, we found that the p-value of the *COL2A1* association strengthened, demonstrating how broadening genetic backgrounds can improve statistical power (p = 1.57x10⁻¹⁵; OR=175.7; *ptv* model) and *VSX2* (p = 3.15x10⁻¹⁰; OR=2.73; *flexnonsynmtr* model). Future studies incorporating sequence data from individuals of diverse ancestries will be crucial for further uncovering the genetic architecture of RD.

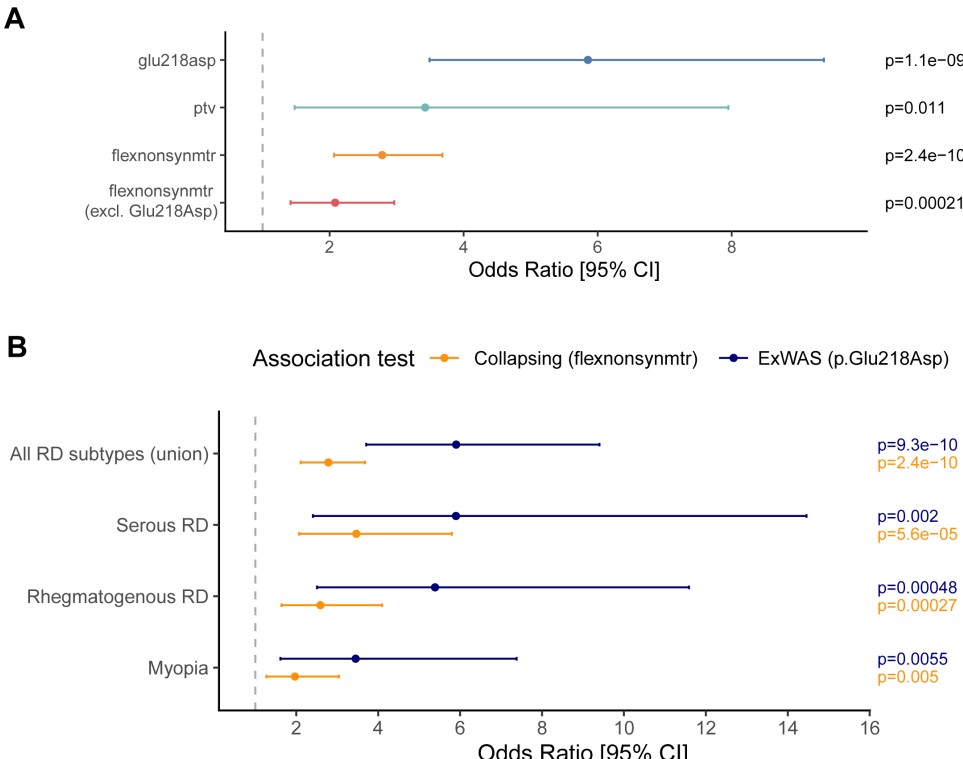

**Fig 3. *VSX2*-retinal detachment associations. (A)** Stratified analysis demonstrating the association between *VSX2* and RD for the **p.**Glu218Asp variant identified in the ExWAS, the gene-level association under the *ptv* and *flexnonsynmtr* collapsing analysis models, and the gene-level association when **p.**Glu218Asp carriers were excluded. **(B)** Association between *VSX2* and RD subtypes based on ICD10 billing codes.

## *VSX2* replication analysis

We next replicated the association between *VSX2* and retinal detachment in two independent, whole-genome-sequenced case-control cohorts: All of Us (AoU) and the 100,000 Genomes Project (100kGP). Amongst European participants, AoU included 987 cases and 35,913 controls, while 100kGP comprised 344 cases and 16,442 controls. We first focused specifically on the c.654G>T (p.Glu218Asp) variant identified in the UKB ExWAS. Despite their smaller sample sizes and reduced statistical power relative to UKB, as well as differences in phenotypic ascertainment, both cohorts yielded effects that were directionally concordant with the UKB association (100kGP: OR = 15.65, p = 0.011; AoU: OR = 5.2, p = 0.067) (**S5 Table**). Combining evidence across all three cohorts via a Cochran-Mantel-Haenszel (CMH) test strengthened support for the *VSX2* association even further (p = 1.08x10⁻¹¹). We then sought to replicate the flexnonsynmtr collapsing analysis signal for *VSX2,* which similarly showed directional concordance with the UKB association (100kGP: p = 0.21, OR= 2.4; AoU: p = 0.21, OR=1.7). Meta-analyzing across all three cohorts again improved the significance of the association (CMH p = 1.40x10⁻¹⁰). Collectively, these data demonstrate an unequivocal association between *VSX2* and a substantial increased risk of RD, particularly for the p.Glu218Asp missense variant.

## *VSX2* association with RD is independent of other risk factors

Because RD frequently co-occurs with other ocular conditions (**S6 Table**), we performed sensitivity analyses to determine whether the *VSX2* association is independent of these risk factors. Myopia is one of the largest risk factors for RD, so we first tested whether *VSX2* was associated with mean spherical equivalent (MSE), an aggregate measure of refractive

error [43,44]. In 115,156 EUR participants with available refractometry data, we observed only nominal associations with decreased MSE for both the p.Glu218Asp variant ($\beta = -0.99$, SE: 0.32, $p = 0.002$) and the gene-level collapsing analysis (*flexnonsynmtr* $\beta = -0.38$, SE: 0.15, $p = 0.01$). To determine whether *VSX2* associates with RD independently of myopia, we repeated the variant- and gene-level association tests adjusting for MSE in the sex-balanced RD case-control cohort with available MSE measurements (61,648 individuals) Despite reduced statistical power, the *VSX2*-RD associations remained significant ($p < 0.05$) after conditioning on MSE for both the gene- and variant-level tests (**S7 Table**).

We next assessed the potential influence of cataract and glaucoma, two other well-established RD risk factors [10]. After excluding cases with a documented history of cataract diagnosis prior to being diagnosed with RD and cases with a history of glaucoma, we observed that the association between *VSX2* and RD remained significant (**S8 Table**). Collectively, these findings indicate that the association between *VSX2* and RD appears to be independent of other known RD risk factors.

## RD subtype analysis

We next tested whether *VSX2* was more strongly associated with subtypes of RD, including rhegmatogenous (ICD10: H33.0; n = 2,753 cases) and serous RD (ICD10: H33.2; n = 1,618 cases). Under the *flexnonsynmtr* model, we observed an association of *VSX2* with both serous RD ($p = 5.6 \times 10^{-5}$; OR [95% CI] = 3.4 [1.9-5.8]) and rhegmatogenous RD ($p = 2.7 \times 10^{-4}$; OR [95% CI] = 2.6 [1.5-4.1]), with a higher OR for serous RD, though the confidence intervals largely overlap (**Fig 3B**). The *VSX2* p.Glu218Asp ExWAS variant showed similar associations with RD subtypes, albeit with wider confidence intervals (**Fig 3B**). It is important to note that previous studies have suggested that the relatively high prevalence of serous RD (~22% of UKB RD cases) compared to epidemiological prevalence estimates may reflect inaccuracies in ICD-10 billing code entries [15]. Among the 54 *VSX2* QV carriers who had any form of RD, 19 (35%) were coded for serous RD. As a comparison, only one of the 10 (10%) *COL2A1* PTV carriers were coded for serous RD. Thus, despite potential billing inaccuracies, these findings suggest that the association between *VSX2* and serous RD may warrant further investigation.

We next queried our prior phenome-wide association study (PheWAS; azphewas.com) [19] to determine whether *VSX2* was associated with any other binary or quantitative phenotypes in the UK Biobank. Among ~18,000 binary phenotypes, RD was the only phenotype associated with *VSX2* at a nominal significance threshold of $p < 1 \times 10^{-4}$. Among quantitative traits, the p.Glu218Asp *VSX2* variant was nominally ($p < 1 \times 10^{-4}$) associated with reduced thickness of the inner nuclear layer-retinal pigment epithelium (INL-RPE) in the central subfield ($p = 9.4 \times 10^{-5}$; $\beta = -0.64$, standard error: 0.16). Although the INL-RPE measurement is often used as a proxy for photoreceptor thickness, it also encompasses interneurons, suggesting potential alterations in photoreceptor structure and/or the inner retina [45]. While no other retinal optical coherence tomography (OCT)-derived measurements achieved a $p < 1 \times 10^{-4}$, there was a consistent trend toward associations with phenotypes suggestive of reduced retinal thickness with *VSX2* variation, particularly for the p.Glu218Asp variant (**S9** and **S10 Tables**). Common variants near *VSX2* have also previously been associated with decreased photoreceptor thickness [37,46]. Interestingly, a case series focused on homozygous microphthalmia cases reported that two heterozygous parents exhibited significant loss of macular layers [38]. Of note, OCT measurements are available for only a limited subset of UKB participants (n = 57,785 EUR with WGS), which likely limited statistical power. While these results need to be validated in larger sample sizes, they suggest that heterozygous variants in *VSX2* could lead to structural alterations in the retina, which in turn predispose to retinal detachment.

## Structural and in-silico analysis of *VSX2* Variants

We next compared the heterozygous *flexnonsynmtr* QVs associated with RD to previously reported pathogenic homozygous *VSX2* variants (**Fig 4A** and **4B**). We manually curated 16 pathogenic homozygous variants (7 missense variants, 9 PTVs). All nine PTVs were associated with microphthalmia/anophthalmia, while five missense variants were associated

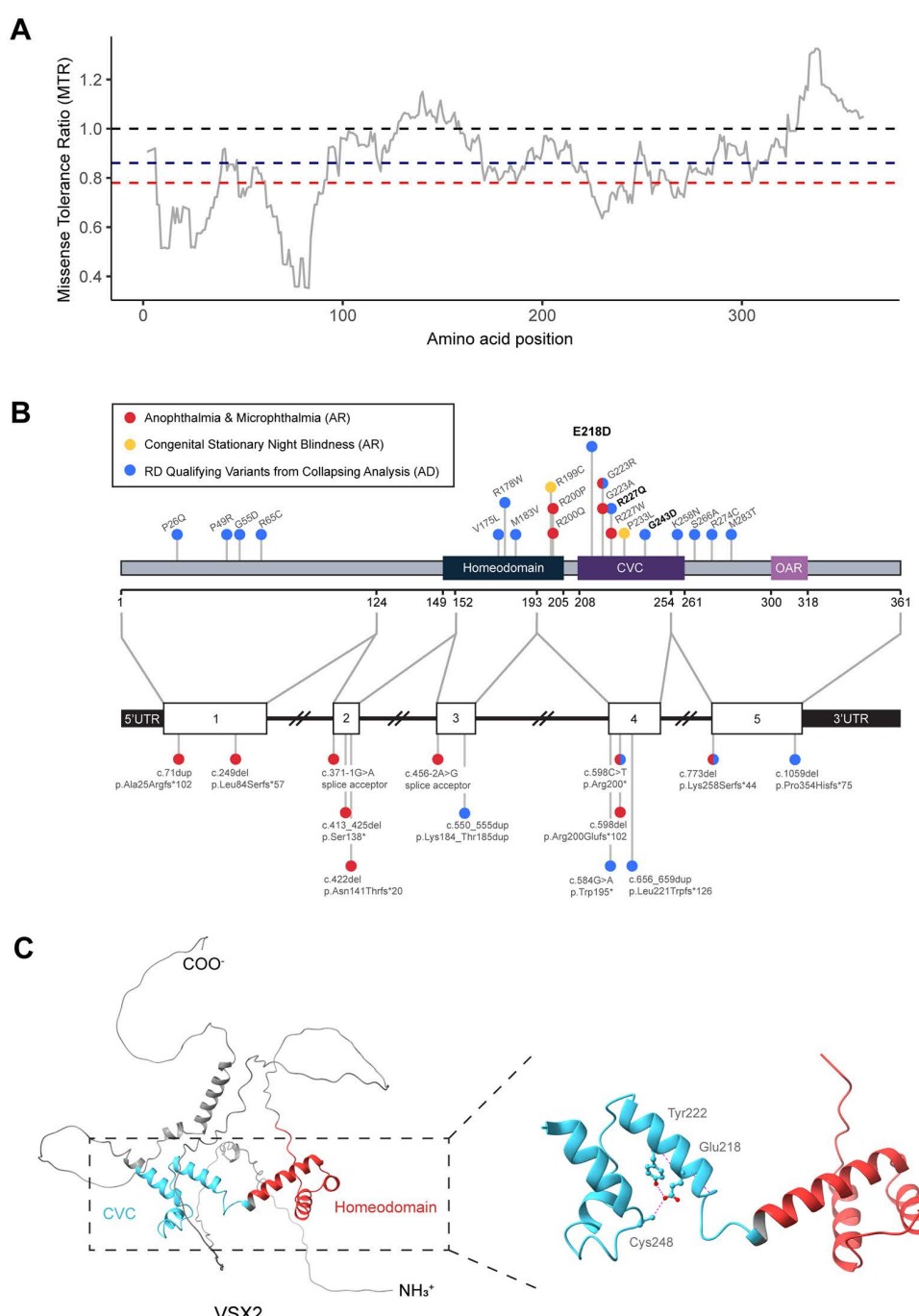

**Fig 4. Structural analysis of *VSX2* Variants. (A)** Missense tolerance ratio (MTR) [42] across the amino acid sequence of *VSX2*. Lower scores correspond to more intolerant regions. Dashed lines represent the intragenic 50th percentile (blue), and the exome-wide 25th percentile (red). **(B)** Lollipop plot illustrating the locations of pathogenic biallelic *VSX2* variants and qualifying variants from the collapsing analysis under the *flexnonsynmtr* model. Bolded variants indicate those that also achieved a p < 0.01 in the ExWAS, including the **p.**Glu218Asp variant. Variants are colored by phenotype, with split coloring indicating multiple reported phenotypes in individual variants. The upper lollipop plot shows missense variants, and the lower plot shows PTVs relative to exon structure (indels, splice-site, premature stop, and frameshift). **(C)** *VSX2* protein structure generated by AlphaFold [47] illustrating predicted hydrogen bond interactions of Glu218 with Tyr222 and Cys248.

with microphthalmia/anophthalmia and two were associated with congenital stationary night blindness (CSNB) (**S11 Table**). In the UK Biobank, there were 21 QVs under the *flexnonsynmtr* model (15 missense variants, 6 PTVs) (**S12 Table**). Notably, three of the heterozygous RD QVs we observed in the UK Biobank sample were previously reported pathogenic homozygous variants associated with microphthalmia/anophthalmia (c.598C > T [p.Arg200*] [48], c.667G > A [p.Gly223Arg] [49], c.773del [p.Lys258Serfs*44] [35]). The pathogenic recessive PTVs and the RD-associated heterozygous PTVs appeared to be spread throughout the gene, indicative of haploinsufficiency [50].

All of the previously reported pathogenic homozygous missense variants clustered in the homeodomain and CVC domain, consistent with these regions being documented as highly intolerant to missense variation as detected by MTR (**Fig 4A** and **4B**). Notably, 53% (8/15) of the qualifying *flexnonsynmtr* missense variants clustered within these same domains. The CVC domain assists the homeodomain in high-affinity DNA binding, contributing to the role of *VSX2* as transcriptional regulator [51,52]. Given its strong effect size, we highlighted protein structural implications for the p.Glu218Asp missense variant. Using AlphaFold to model protein folding, Glu218 was predicted to form hydrogen bonds with Tyr222 and Cys248 within an alpha helix of the CVC domain (**Fig 4C**). The p.Glu218Asp missense variant may potentially disrupt these interactions and alter DNA-binding activity.

To assess the impact of the p.Glu218Asp (E218D), we overexpressed wild-type (WT) *VSX2*, the E218D variant, or the microphthalmia-associated p.Arg227Trp (R227W) variant in ARPE-19 cells, which do not endogenously express *VSX2*. While WT *VSX2* induced a robust increase in cell size relative to control (p = 1.8x10$^{-22}$), the R227W variant showed no effect (p = 0.17; **S3A** and **S3B Fig**). The E218D variant displayed an intermediate phenotype, increasing cell size (p = 6.1x10$^{-13}$) but less than WT (p = 0.009). Principal component analysis of RNA-seq data showed clear transcriptional separation among cells expressing the three forms of *VSX2* (**S3C Fig**). Differential expression analysis identified 819 differentially expressed genes (DEGs) in WT *VSX2* versus control cells, 354 DEGs in R227W versus WT, and 86 DEGs in E218D versus WT (**S3D Fig** and **S13**-**S18 Tables**).

Consistent with its role as direct transcriptional repressor of the gene *MITF,* WT *VSX2* strongly reduced *MITF* expression relative to control (3.95-fold, p = 2.7x10$^{-73}$; **S3E Fig**), as did the E218D variant (4.0-fold decrease; p = 7.1x10$^{-74}$). In contrast, the R227W variant, which has been previously shown to impair DNA binding [51], exhibited weaker repression compared to WT and E218D (1.79-fold reduction; R227W versus WT p = 6.7x10$^{-23}$). Analysis of WNT signaling targets revealed similar allele-specific effects (**S3F Fig**). In line with known repressive effect of *VSX2* on the WNT pathway [24], WT *VSX2* robustly downregulated *WNT2B* and maintained high expression of the WNT antagonist *DKK1*. R227W failed to elicit these responses, whereas E218D displayed an intermediate phenotype, indicating a graded loss of WNT regulation among the variants. Together, these findings indicate that while E218D retains full repressive capacity against *MITF*, it exhibits a selective impairment in regulating cell morphology and WNT signaling, distinguishing its molecular profile from the broad defects observed in R227W.

## *VSX2* is expressed in bipolar cells and Müller glia

*VSX2* plays an important role in regulating bipolar cell fate during retinal development, but its function in the adult retina is less well understood [53,54]. Analysis of a publicly available single-cell RNA-sequencing dataset from adult human retina samples [55] revealed *VSX2* expression in both retinal bipolar cells and some Müller glia (**Fig 5A**). While *VSX2* is a known marker of Müller glia, its specific role in these cells, particularly following injury, remains unclear [56,57]. However, given Müller glia's central role in providing structural support, maintaining retinal homeostasis, and regulating the blood-retinal barrier, dysregulation of *VSX2* in these cells may contribute to the pathogenesis of RD [58]. Alternatively, *VSX2* dysregulation may disrupt bipolar cell-cell adhesion proteins, a process implicated in retinoschisis [59] (see Discussion). Further functional work will be required to elucidate the role of Müller glia or other cell types in *VSX2*-mediated RD.

Given the diversity of bipolar cell function in the adult retina and the numerous distinct bipolar cell clusters, we also investigated *VSX2* expression patterns more closely in bipolar cell subtypes (**Fig 5B**). Of these subclasses, *VSX2*

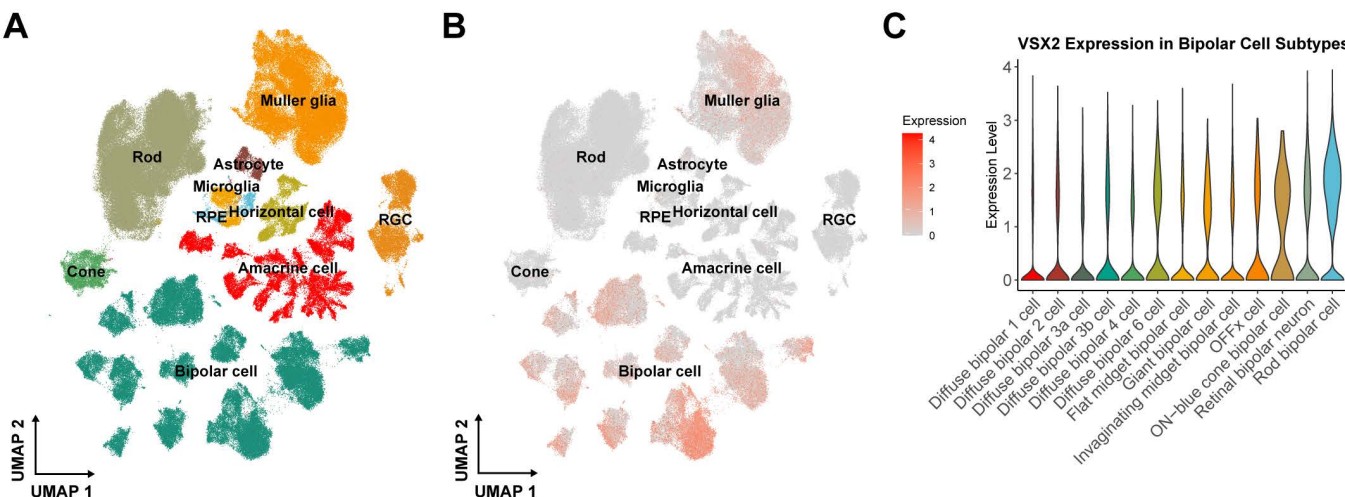

**Fig 5. Cell-specific expression of *VSX2* in the adult retina. A)** UMAP of scRNA-seq data demonstrating retinal cell types and **B)** expression of *VSX2* in bipolar neurons and Müller glia cells. **C)** Expression of *VSX2* in 13 distinct bipolar cell subtypes.

expression was relatively lower in diffuse bipolar cell subtypes, which are involved in processing of visuospatial information in light conditions (Fig 5C) [60]. In contrast, *VSX2* expression was highest in rod bipolar cells, which process and amplify signals from highly light-sensitive rod photoreceptors thus permitting low-light vision [60]. In a fetal single-cell dataset (S4 Fig) [61], *VSX2* exhibited high expression in bipolar cells and Müller glia, mirroring the adult pattern. However, *VSX2* is also expressed in fetal retinal progenitor cells, highlighting its role in retinal development and cell differentiation.

## Top-ranking collapsing analysis genes are broadly enriched for recessive genes

The phenotypic expansion of *VSX2* reported in this article highlights a spectrum of ocular abnormalities, from severe developmental defects like microphthalmia and anophthalmia in homozygotes, to increased RD risk in heterozygous carriers, demonstrating how dosage of this gene influences phenotypic severity (Fig 6A). Notably, early-onset RD has been reported in *VSX2*-related microphthalmia [29]. This is consistent with the emerging paradigm that heterozygous carriers for many recessive disease genes may be at increased risk of attenuated, yet related phenotypes [22,23].

We next investigated whether the *VSX2* finding represents a broader trend of gene dosage effects across Mendelian ocular disease genes. We examined whether the collection of nominally significant genes (p < 0.05; n = 3,064 genes) found across our 10 collapsing analysis models was enriched for both dominant and recessive ocular disease genes (n = 847 qualifying ocular disease genes from Genomics England). After excluding Stickler syndrome-associated genes, which have well-established associations with RD, we tested whether this overlap exceeded random expectation using Fisher's exact test (Methods). 117 known autosomal recessive ocular disease risk genes overlapped with nominally associated RD genes from our collapsing analyses (p = 1.65x10$^{-3}$; OR [95% CI] = 1.45 [1.16-1.79]) (Fig 6B and 6C and S19 Table). One example is the *LTBP2*, a gene for which recessive variants are linked with congenital glaucoma and microspherophakia [62] (UK Biobank RD association p = 4.4x10$^{-5}$; OR [95% CI] = 3.65 [2.14-6.22]; *flexdmg* model). *LOXL3* is another example, in which recessive mutations have been linked to severe myopia and early-onset cataracts and retinal detachment (UK Biobank RD p = 2.4x10$^{-4}$, OR [95% CI] = 3.65 [2.01-6.63]; *ptv* model) [63]. Likewise, recessive mutations in *P3H2* cause severe myopia, cataracts, vitreoretinal degeneration, and childhood-onset RD (UK Biobank RD p = 0.0017; OR [95% CI] = 1.59 [1.21-2.09]; *flexdmg* model) [64]. The potential implications for this overlap between RD and recessive ocular disease genes are widespread, as Hanany et al. [65] previously demonstrated that heterozygous carriers of

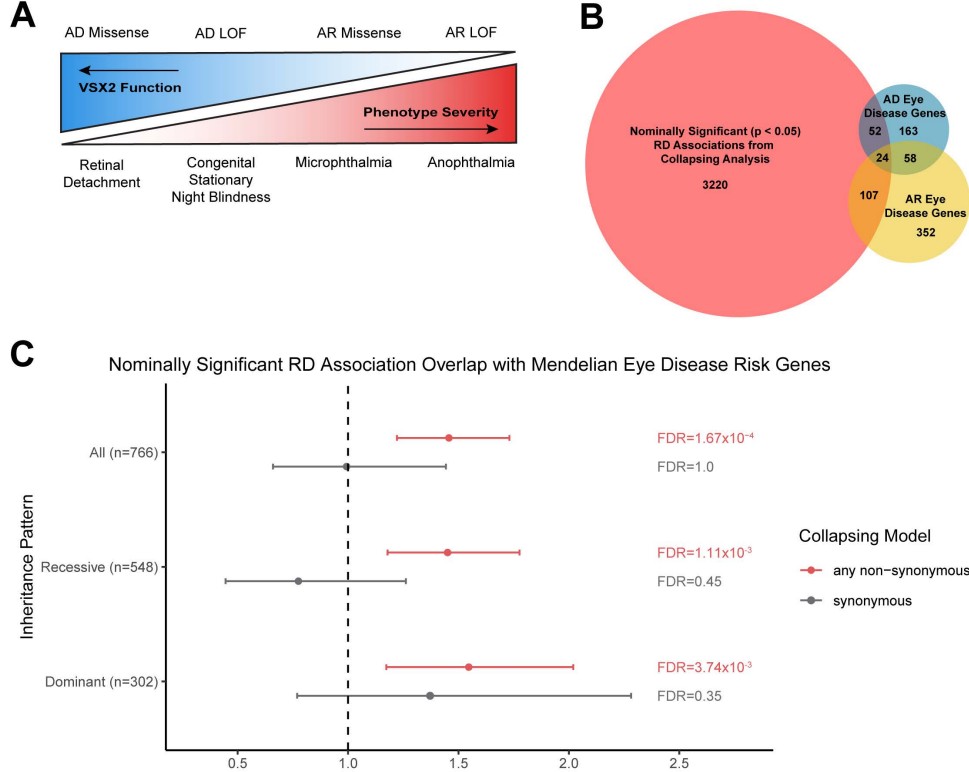

**Fig 6. Phenotypic spectrum model for *VSX2*-associated pathologies and overrepresentation analysis. (A)** Working simplified model of the spectrum of genotype-phenotype associations driven by changes in *VSX2* dosage. **(B)** Overrepresentation analysis for 3,064 genes nominally associated with RD in the collapsing analysis (p<0.05) and Mendelian eye disease risk genes from the Genomics England PanelApp. Documented eye disease genes were split according to inheritance pattern (autosomal recessive=AR, autosomal dominant=AD). Overlapping genes can be found in S19 Table. **(C)** Forest plot depicting significant overrepresentation of genes nominally associated with RD derived from the collapsing analysis and groups of known eye disease risk genes.

autosomal recessive retinal diseases may exist at rates as high as 1 in 2.26 individuals of European ancestry – among the highest carrier frequencies for any group of Mendelian conditions.

There was a similar enrichment of autosomal dominant ocular disease genes among the nominally significant collapsing analysis hits (p=1.65x10⁻³; OR [95% CI] = 1.61 [1.21-2.12]). This list includes *SIX3,* which is associated with a severe Mendelian phenotype characterized by holoprosencephaly and ocular manifestations including microcornea, coloboma, and others (UK Biobank RD p=1.1x10⁻³, OR [95% CI] = 3.81 [1.91-7.60]; *raredmgmtr* model) [66]. Another example is *NHS*, in which PTVs cause Nance-Horan syndrome, characterized by congenital cataracts, dysmorphic features, and variable intellectual disability [67]. This gene was nominally associated with RD in the UK Biobank under the *flexnonsynmtr* model (p=2.7x10⁻³, OR [95% CI] = 1.6 [1.2-2.1]). One possible explanation for these signals is that the RD associations are driven by milder variants (e.g., hypomorphic missense) than those causing more severe phenotypes within the same gene. Collectively, these results demonstrate a significant overlap between the genetic architecture of Mendelian eye disease genes and adult-onset RD.

## Discussion

To our knowledge, this study represents the largest sequence-based analysis of retinal detachment to date, encompassing 7,276 cases and 236,741 controls with WGS. Our variant- and gene-level analyses uncover *VSX2* as a novel genetic

determinant of RD risk while validating previously established associations including *FAT3*, *RDH5*, and *COL2A1*. We found that rare heterozygous missense variants in *VSX2* confer up to a 6-fold increased risk of RD, representing a phenotype expansion distinct from severe developmental disorders observed in recessive carriers of *VSX2* mutations. This discovery of *VSX2* as an RD risk gene exemplifies how monoallelic carriers of recessive disease alleles can manifest attenuated phenotypes. Moreover, our findings extended to other genes associated with Mendelian disorders, demonstrating that genes traditionally associated with severe developmental eye disorders may contribute to RD risk through heterozygous variants. Collectively, this work advances our understanding of the genetic architecture of RD and motivates a broader paradigm of gene dosage effects in ophthalmological disorders.

*VSX2* encodes the VSX2 homeobox protein, a transcription factor that acts largely as transcriptional repressor in eye organogenesis [51]. The protein contains several key domains, including the highly conserved homeobox (HOX) and CVC domains [38,39,61]. Biallelic variants in the CVC and HOX domains of *VSX2* have been linked to conditions such as microphthalmia, anophthalmia, and more recently, congenital stationary night blindness [29,39,48,49]. The RD-associated variants identified in our analyses here are dispersed across *VSX2*, and the p.Glu218Asp variant specifically lies within the CVC domain (**Fig 4B**). Mouse studies of *Vsx2* indicate that the CVC domain supports the HOX domain in high affinity DNA-binding, although the precise mechanism remains unresolved [67]. Given the enrichment of pathogenic variation in the CVC domain, p.Glu218Asp variant may alter VSX2's DNA binding activity or regulatory specificity. Notably, this p.Glu-218Asp variant has not been previously associated with microphthalmia, and it remains unclear whether homozygous individuals for this variant will present with microphthalmia. To further investigate how p.Glu218Asp functionally differs from microphthalmia alleles, we performed *in vitro* assays comparing its transcriptional regulatory effects to wildtype *VSX2* and the well-characterized R227W variant.

During retinogenesis, VSX2 is essential for retinal progenitor cell proliferation and for bipolar cell differentiation [26,51,68]. A key function by which VSX2 promotes retinal progenitor specification is through repression of WNT pathway genes and direct inhibitory binding of *MITF*, which collectively promote neural retinal identity at the expense of RPE fate [24]. In mice, disruption of *Vsx2* results in a thinner neural retina and thickened RPE due to inappropriate Mitf regulation, which has an established role in RPE differentiation [51,69]. In our functional assays, overexpression of WT *VSX2* in RPE cells, which do not endogenously express the gene, reduced *MITF* expression as expected. In contrast, the R227W exhibited impaired repression of *MITF* and dysregulation of WNT-related genes, which aligns with previous reports of impaired DNA-binding activity and broad disruption of VSX2-mediated transcriptional regulation [51]. Crucially, E218D retained its repressive capacity against *MITF* but failed to fully regulate WNT pathway genes. E218D produced intermediate effects on *WNT2B* and *DKK1* expression, as well as on RPE morphology, falling between WT and R227W. This selective functional impact suggests that E218D produces a milder yet detectable perturbation of VSX2 regulatory activity compared to R227W, consistent with its association with the less severe phenotype of isolated RD. However, these findings must be interpreted in the context of important experimental constraints. Because ARPE-19 cells do not endogenously express *VSX2,* our overexpression assays assess the variant in isolation and cannot capture potential dominant-negative variant effects. Therefore, future studies will require introducing the E218D allele into Muller glia or bipolar cells, where *VSX2* is naturally expressed in the adult retina.

*VSX2* expression persists in adulthood in both bipolar cells and Müller glia. In mature bipolar cells, *VSX2* is not essential for maintaining cell identity but may support physiological function [70,71]. Disrupted bipolar cell-cell adhesion has been hypothesized to contribute to retinoschisis in the inner nuclear and outer plexiform layers, a structural defect which can be misdiagnosed as RD [59,72]. Consistent with this hypothesis of compromised structural integrity, we observed an association of reduced thickness of the INL-RPE in E218D carriers. As the measurement encompasses the bipolar-containing inner nuclear layer, this thinning may reflect subtle atrophy that predisposes the retina to detachment. Alternatively, dysfunctional Müller glia may contribute to RD susceptibility. Müller glia regulate ion homeostasis, maintain the blood-retinal barrier, and coordinate response to retinal stress/injury [73]. Aberrant Müller glia responses to stress and

aging could weaken retinal structural integrity or the blood retinal barrier, providing a possible mechanism by which partial VSX2 dysfunction increases RD risk. Nevertheless, the physiological role of *VSX2* in the mature retina, and how heterozygous variants such as p.Glu218Asp alter its function, remains an important direction for future study.

Recent studies have challenged the traditional view that heterozygous carriers of recessive Mendelian diseases are asymptomatic [22,23,74]. In addition to ophthalmologic disorders, emerging research has documented similar complex genetic interactions across various conditions. For instance, heterozygous carriers of *CFTR* have been found to have increased risks of bronchiectasis, pancreatitis, and diabetes [75]. In the context of eye disorders, our study provides a compelling example by demonstrating that heterozygous *VSX2* carriers are at increased risk of RD. A notable case series by Iseri et al. further supports this complexity, describing a consanguineous Iranian family where heterozygous parents of children with biallelic *VSX2* variants exhibited inner retinal dysfunction, subnormal rod and cone electroretinogram results, and decreased retinal thickness on OCT, despite being clinically asymptomatic [48]. This heterozygous carrier phenotype suggests a semidominant effect and supports our proposed model of *VSX2* variants existing on a broader phenotypic spectrum. Furthermore, for several other recessive ocular disease genes such as *LTBP2, LOXL3,* and *P3H2,* our findings support a broader pattern in ocular genetics in which heterozygous variants contribute to milder or later-onset phenotypes like RD. Collectively, these observations reinforce a gene-dosage spectrum that likely extends beyond *VSX2* and may have larger implications for interpreting genetic risk factors for complex eye diseases [23].

Our study has several limitations. Primarily, billing records for different retinal detachment subtypes are likely imprecise, making it difficult to fully understand the phenotypic presentation of *VSX2*-mediated retinal detachment. Careful phenotyping will need to be performed in future studies to better understand the mechanisms behind this association. Additionally, despite our efforts to maximize genetic diversity representation in our analyses, the relative lack of demographic diversity in the UK Biobank limits our ability to understand population-specific variation of *VSX2* and its phenotypic associations. For example, the vast majority of *VSX2* recessive disease variants have been described in consanguineous populations of Middle Eastern genetic ancestry, a group that is underrepresented in all biobanks in this study. Greater intentional efforts to generate biobank medical research resources from populations of non-European ancestries are essential for a more comprehensive understanding of *VSX2*-related pathologies. Additionally, larger familial genetic collections of parents with children affected by *VSX2*-associated microphthalmia and anophthalmia should be conducted to assess their risk for RD. Future studies in cellular and rodent models will be critical to elucidate the pathophysiology of *VSX2*-induced RD.

In this study, we leveraged population-scale whole-genome sequencing data to identify a novel association between heterozygous non-synonymous variants in *VSX2* and increased risk of retinal detachment. Our analyses of *VSX2* protein structure and *VSX2* single-cell expression patterns, coupled with *in vitro* assays of *VSX2* variants in RPE cells, illuminate potential mechanisms underlying this association. Furthermore, by extending our *VSX2* findings, we revealed a significant overlap between genes associated with Mendelian conditions and those linked to RD risk in the UK Biobank population sample. These discoveries not only expand the phenotypic spectrum of *VSX2*-related pathologies but also highlight the crucial role of rare variants and their dosage effects in understanding the pathogenesis of complex traits like RD. Our findings establish a framework for investigating similar genetic mechanisms in other ophthalmological disorders and emphasize the value of comprehensive genomic analyses of biobank data in uncovering novel disease associations.

## Materials and methods

### UK Biobank

**Ethics statement.** The UKB is a prospective study of approximately 500,000 participants aged 40–69 years at the time of recruitment. Participants were recruited in the United Kingdom between 2006 and 2010 and are continuously followed. Participant data include health records that are periodically updated by the UKB, self-reported survey information, linkage to death and cancer registries, collection of urine and blood biomarkers, imaging data, accelerometer data, and various

other phenotypic endpoints. All study participants provided written informed consent, and the UKB has approval from the North-West Multi-centre Research Ethics Committee (11/NW/0382).

**Phenotype data.** The initial cohort consisted of 490,560 multi-ancestry cases from the UK Biobank. We parsed phenotypic data using our publicly available R package, PEACOCK (https://github.com/astrazeneca-cgr-publications/PEACOK). In addition, as previously described, we grouped relevant ICD-10 codes, death registry data, and self-reported data into clinically meaningful "Union" phenotypes [19]. The control cohort was restricted to exclude individuals with a history of eye disease, using ICD-10 Chater VII codes (diseases of the eye and adnexa).

Retinal detachment (RD) was our primary phenotype of interest. The "Union" RD phenotype, we included ICD-10 code H33.0 and all of its sub-codes. For RD subtype analyses, we considered Union H33.2 (serous RD), Union H33.0 (retinal detachment with retinal break; i.e., rhegmatogenous RD). We excluded Union H33.4 (traction RD) due to its small sample size (n = 158 European cases).

We also accessed refractometry data that was available for 115,156 of the European participants with available whole-genome sequencing data. Using these data, we calculated mean spherical equivalent (MSE) values per participant as previously described [76]. Briefly, we first calculated mean spherical power (UKB data fields 5084 and 5085) and mean cylindrical power (data fields 5086 and 5087) for each eye. We then calculated MSE refractive error as Spherical Power + (0.5 × Cylindrical Power). We then took the mean MSE per eye.

**WGS data.** Whole-genome sequencing (WGS) data of the UKB participants were generated by deCODE Genetics and the Wellcome Trust Sanger Institute as part of a public-private partnership involving AstraZeneca, Amgen, GlaxoSmithKline, Johnson & Johnson, Wellcome Trust Sanger, UK Research and Innovation, and the UKB. The WGS sequencing methods have been previously described [77,78] (p150). Briefly, genomic DNA underwent paired-end sequencing on Illumina NovaSeq6000 instruments with a read length of 2x151 and an average coverage of 32.5x. Conversion of sequencing data in BCL format to FASTQ format and the assignments of paired-end sequence reads to samples were based on 10-base barcodes, using bcl2fastq v2.19.0. Initial quality control was performed by deCODE and Wellcome Sanger, which included sex discordance, contamination, unresolved duplicate sequences, and discordance with microarray genotyping data checks.

The 490,560 UK Biobank genomes were processed at AstraZeneca using the provided CRAM format files. A custom-built Amazon Web Services cloud compute platform running Illumina DRAGEN Bio-IT Platform Germline Pipeline v3.7.8 was used to align the reads to the GRCh38 genome reference and to call small variants. Variants were annotated against the transcript for which they are most damaging using SnpEff v4.3 [79] and Ensembl Build 38.92 [80].

**Sample QC.** Sample-level filters were applied as previously described to derive a subset of UKB suitable for analysis [19]. We removed any person withdrawn from UKB and without linked WGS data. Where available, WGS data were checked for concordance with previous exome sequencing and genotyping array data releases using the KING relatedness software v2.2.3 [81]. KING kinship coefficients are scaled such that duplicate samples have kinship 0.5. We thus removed samples with kinship <0.49 to exome data or <0.465 to array data – empirically derived thresholds that account for platform-specific variation and reliably identify mismatches and sample swaps [20].

Of the remaining samples, we retained only those with VerifyBamID FREEMIX contamination <0.04 and ≥10x coverage across ≥94.5% of Consensus Coding Sequence (CCDS) database (release 22) bases. The cohort was filtered further to select the largest subset of individuals with ≤0.3536 kinship estimate from KING, removing any first-degree relatives.

We inferred ancestry across four superpopulation cohorts: African (AFR), East Asian (EAS), European (EUR), and South Asian (SAS) based on the 1000 Genomes phase 3 cohort [82] using the Peddy (v0.4.2) software package [83]. We retained people with ≥0.90 probability of belonging to an ancestry group. For EUR ancestry, which represents >90% of the dataset, we further restricted to people ≤4 standard deviations (SD) from the mean of the first four genetic principal components (PCs). Female controls were pseudo-randomly down sampled if there was a significant difference in the odds of being male across cases and controls from a given ancestry cohort (Fisher's Exact Test p-value <0.05). The final Union

H33 cohort included 7,276 individuals of European ancestry and 7,593 individuals in the multi-ancestry group. The control group comprised 236,741 European ancestry individuals and 251,496 multi-ancestry individuals.

## Exome-wide association study (ExWAS)

We performed variant-level association tests for the binary and quantitative traits. We included protein-coding variants identified in at least six individuals from the predominantly unrelated European ancestry UKB genomes as previously described [19]. Variants were required to meet the following QC metrics for inclusion in the test: coverage ≥ 10x; ≥ 0.20 of reads are for the alternate allele for heterozygous genotype calls; binomial test of alternate allele proportion departure from 50% in heterozygous state $p ≥ 1x10^{-6}$; GQ ≥ 20; Fisher Strand Bias (FS) ≤200 for indels and ≤60 for SNVs; root mean square mapping quality (MQ) ≥40; QUAL ≥30; read position rank sum score (RPRS) ≥−2; mapping quality rank score (MQRS) ≥−8; DRAGEN variant status = PASS; ≤ 10% of the cohort have missing genotypes. Additional out-of-sample QC filters were also imposed based on the gnomAD v2.1.1 exomes (GRCh38 liftover) dataset [40]. The site of all variants should have ≥ 10x coverage in ≥30% of gnomAD exomes and, if present, the variant should have an allele count ≥50% of the raw allele count (before removing low-quality genotypes) in the dataset. Variants with missing values on a particular filter were treated as passing that filter, and therefore retained unless failing another metric. Those present in fewer than 6 people from UKB EUR ancestry population, or failing QC in >20,000 people, were also removed. P values were generated by adopting Fisher's exact two-sided test. Three distinct genetic models were studied for binary traits: allelic (A versus B allele), dominant (AA + AB versus BB), and recessive (AA versus AB + BB), where A denotes the alternative allele and B denotes the reference allele. For quantitative traits, we adopted a linear regression (correcting for age and sex) and replaced the allelic model with a genotypic (AA versus AB versus BB) test.

## Collapsing analysis

We performed collapsing analyses as previously described [19]. Briefly, we defined 10 collapsing analysis QV models to test gene-level associations with different types of coding-sequence variation, including a synonymous variant model as an empirical negative control. Per-model filters for selecting qualifying variants are described in **S2 Table**. The following in-sample variant QC metrics were applied universally across the collapsing analysis models: coverage ≥ 10x; present in CCDS (release 22) [84]; ≥ 0.8 of reads are for the alternate allele among homozygous genotype calls; [0.25,0.8] of reads are for the alternate allele for heterozygous genotype calls; binomial test of alternate allele proportion departure from 50% in heterozygous state $p ≥ 1x10^{-6}$; GQ ≥ 20; FS ≤ 200 for indels and ≤60 for SNVs; MQ ≥ 40; QUAL ≥30; RPRS ≥−2; MQRS ≥−8; DRAGEN variant status = PASS. Additional out-of-sample QC filters were also imposed based on the gnomAD v2.1.1 exomes (GRCh38 liftover) dataset [40]. For all variants, the site should have ≥ 10x coverage in ≥25% of gnomAD exomes. For variants present in gnomAD, we retained those with RPRS ≥ -2 and MQ ≥ 30 in that cohort. Variants with missing values on a particular filter were treated as passing that filter and retained unless failing another metric.

For binary traits, the difference in the proportion of cases and controls carrying QVs in a gene was tested using Fisher's exact two-sided test. For quantitative traits, the difference in mean between the carriers and noncarriers of QVs was determined by fitting a linear regression model, correcting for age and sex. Based on our previously published n-of-1 permutation analysis and synonymous model p-value distribution, we set a conservative p-value cutoff of $p < 1x10^{-8}$ to define significant associations [19].

We focused on the European subset as the discovery cohort, given the much larger sample size (>90% of UKB participants). However, we also performed the identical collapsing analysis in the South Asian, East Asian, and African UKB cohorts, focusing on the Union H33 phenotype. We then performed a pan-ancestry analysis, combining the results from these three cohorts and the European cohort using our previously introduced approach of applying a CMH (Cochran Mantel Haenszel) test to generate combined 2 × 2 × N stratified P values, with N representing up to all four genetic ancestry groups.

## Replication study in all of Us

At the time that this study was conducted, All of Us contained short-read WGS data from 245,388 individuals [85]. Full details on genome sequencing and downstream quality control have been previously published [42]. Briefly, PCR-free Barcoded WGS libraries on the Illumina NovaSeq 6000. Following demultiplexing, initial QC analysis was conducted using the Illumina DRAGEN pipeline. All of Us samples passing the following quality control metrics were used in joint calling and released to the research community: mean coverage ≥30x, genome coverage ≥90% at 20x, All of Us Hereditary Disease Risk gene coverage ≥95% at 20x, aligned Q30 bases $\geq 8 \times 10^{10}$, and contamination ≤1%. Additional QC checks included fingerprint mismatch and array mismatch.

To generate a case-control cohort, we defined RD cases as those who were billed with the corresponding RD-related ICD10 codes (H33.0, H33.1, H33.2, H33.4 and H33.8) and ICD9 codes (361.0, 361.1, 361.2, 361.8 and 361.9). To construct the control cohort, we excluded all samples with any history of eye disease based on ICD10 codes H00-H59 and ICD9 codes 360–379.

AoU provides genetic ancestry predictions based on a random forest classifier. We restricted our analysis to individuals of European ancestry (probability of European genetic ancestry ≥ 0.95) and further restricted to samples whose first four principal components were within ±4 s.d. across the top four principal component means (n = 73,288). The dataset was further restricted to include only those with corresponding sex at birth and genetically-inferred sex, resulting in 71,106 samples. Finally, we identified pairs of related individuals using the relatedness.tsv file provided by AoU, which lists all pairs of samples with a kinship score above 0.2 (calculated via the pc_relate function in Hail). We removed the first sample of each related pair, leaving a final cohort of 69,120 unrelated individuals of European ancestry.

AoU participants show a wide distribution of ages compared to UKB, and since aging is a major risk factor in RD, we filtered out 21,448 controls with age less than 45 from the control cohort. This threshold was chosen according to the density plot of ages in cases and controls (S5 Fig). After performing down-sampling on controls to match the sex ratio observed in cases, the final European case-control cohort included 987 cases and 35,913 controls.

We annotated variants in *VSX2* using SnpEff v4.3. We used the *flexnonsynmtr* model to perform gene level collapsing analysis. For a variant to be included, it needed to have either "." or "PASS" in the FILTER fields of both AoU and gnomAD datasets. For variants exclusively in AoU, they were only retained if their FILTER field contained either a "." or "PASS". The missense tolerance ratio (MTR) of variants in the *flexnonsynmtr* model needed to satisfy MTR ≤ 25th %ile or intragenic MTR ≤ 50th %ile. Variants also need to satisfy rf_tp_probability ≥ 0.1 (SNVs), rf_tp_probability ≥ 0.02 (indels). Finally, minor allele frequency (MAF) needed to satisfy MAF ≤ 0.001 within the AoU cohort and within gnomAD. Having extracted qualifying variants satisfying the *flexnonsynmtr* model, we performed Fisher's exact two-sided test on the number of *VSX2* heterozygous carriers among cases and controls.

## 100kGP

Whole-genome sequencing data were generated, as previously described, using TruSeq DNA polymerase-chain-reaction (PCR)–free sample preparation kit (Illumina) on the HiSeq2500 platform. Reads were aligned using the Isaac Genome Alignment Software, and the Platypus variant caller was used for small variant calling [86]. Variants were annotated using VEP (v105) with the gnomAD plugin included [87]. We imposed the following inclusion criteria: VerifyBamID FREEMIX contamination ≤0.03; aligned reads have ≥ 15x coverage across 95% of the genome with MQ > 10; > 90% concordance between variant calls from sequencing and matched genotyping array; median fragment size >250 bp; < 5% chimeric reads; > 60% mapped reads; < 10% AT dropout; concordance between self-reported and genetically determined sex.

Ancestry was inferred by training a random forest model upon PCs 1–8 from the 1000 Genomes phase 3 cohort [82]. People with >99% probability of European ancestry and <4SD from the mean of PCs 1–4 in the 100kGP inferred-European cohort were retained for analysis. Finally, we removed one from each pair of individuals estimated to have ≥2nd degree relatedness (kinship coefficient >0.0442) under the plink2 implementation of the KING-relatedness software. Here

we followed a 3-step procedure, removing one from each pair of related cases with the *ukb_gen_samples_to_remove* R function*,* removing controls related to remaining cases, and finally using *ukb_gen_samples_to_remove* to removing one from each related pair of remaining controls*.* Sex-rebalancing was then performed as for the UK Biobank dataset so that the odds of being male was comparable in cases and controls.

For replicating the *VSX2* ExWAS and flexnonsynmtr signals, we retained variants with the following metrics: missingness ≤0.05; median coverage ≥ 10x; Median GQ ≥ 15; ≥ 0.25 heterozygous allele calls do not show significant (p < 0.01) allele imbalance in binomial test; ≥ 0.50 complete genotype data for a variant; Hardy-Weinberg equilibrium mid p > $1 \times 10^{-5}$ among unrelated samples of European ancestry. Variants were mapped to consequences based on their impact upon the Matched Annotation from NCBI and EMBL-EBI (MANE) transcript (ENST00000261980).

## Structural and *in-silico* variant prediction

A lollipop plot was generated to visualize all RD-associated *VSX2* missense variants using the lollipops software [88]. Structural predictions for *VSX2* folding and amino acid interactions were conducted using AlphaFold [47] and ChimeraX [89–91].

## *In-vitro* functional validation of *VSX2* variants

To assess the function of the lead p.Glu218Asp variant identified in our ExWAS analysis, we overexpressed wild-type (WT) and mutant forms of *VSX2* in human ARPE-19 cells using lentiviral transduction. ARPE-19 cells (ATCC CRL-2302) were cultured in DMEM supplemented with 10% fetal bovine serum and 1% penicillin-streptomycin and maintained at passages <P10. ARPE-19 cells express RPE-associated transcription factors, including *MITF*, yet do not endogenously express *VSX2*, providing a suitable system to assay *VSX2* transcriptional activity [92,93]. *VSX2* lentiviral plasmids (EF1α-VSX2-WPRE-CMV-eGFP), based on prior work from Livne-bar et al. [94], were obtained for the WT sequence (VectorBuilder). Site-directed mutagenesis was used to generate *VSX2* variants, including the p.Glu218Asp variant identified in our study. A known pathogenic variant linked with severe recessive microphthalmia/anophthalmia, p.Arg227Trp, was used as our positive control because it was located in the CVC domain [95]. For our negative transduction control, we used a lentivirus expressing eGFP and mCherry. All plasmids were sequence-verified (Plasmidsaurus) and visualized using IGV prior to expansion in E. coli and plasmid midiprep isolation.

Lentiviruses were produced in HEK-293T cells using polyethylenimine transfection, and viral supernatants were collected for transduction of ARPE-19 cells. Cells were expanded for one week post-transduction, yielding ~20–30% eGFP-positive cells, which were subsequently purified using fluorescence-activated cell sorting (BD FACSAria II). Fifty thousand eGFP$^+$ cells were expanded for an additional week prior to phenotypic assessment. Cellular morphology was evaluated two weeks post-transduction using brightfield and fluorescence microscopy. Cell area (μm$^2$) was measured using ImageJ by manual tracing of 40–60 cells per image from three independent images per experimental condition. Pairwise Wilcoxon test with Bonferroni-adjusted p-values was used to determine statistical significance between conditions.

Total RNA was isolated using the Qiagen RNeasy Micro Kit (74004) and submitted for bulk RNA sequencing (Plasmidsaurus). FASTQ files were aligned to the GRCh38 human reference genome and differential expression analysis was performed using DESeq2 (v1.42.1) in R [96]. Differential expression was defined using an absolute log2(fold change) ≥ 1.5 and a Benjamini-Hochberg-adjusted p-value < 0.05.

## Single-cell RNA-sequencing analysis

Single-cell RNA (scRNA) expression data was downloaded from the Human Cell Atlas, generated from adult retina samples [55]. We first analyzed scRNA sequencing data from 265,767 cells from six healthy adult retinas, as described in the original publication. Using the same cell identity annotations from the original publication, we first examined *VSX2*

expression levels across major cell classes using the R package Seurat version 5.1.0 [10,14]. Additionally, we looked at the expression of RD-associated genes, both from the collapsing analysis and as described in prior publications [52], across major retinal cell classes. To gain better insight into *VSX2* expression in bipolar cell sub-classes, we used scRNA sequencing data of 72,778 bipolar cells from the same cell atlas and retained the original annotations. To examine *VSX2* expression in the developing retina, we downloaded single-nuclear RNA sequencing data of 226,506 cells generated from 14 fetal samples and retained the original annotations [61].

### Overrepresentation Analysis of Nominally Significant RD-Associated Genes

We tested for overrepresentation of nominally significant ($p < 0.05$) RD-associated genes identified in the collapsing analysis with documented ocular disease risk genes collected from the Genomics England PanelApp [48]. We collated 847 unique genes across all available ocular panels (n = 22). The ptv, URmtr, flexnosynmtr, UR, flexdmg, rec, raredmg, ptvraredmg, and raredmgmtr models were used to identify nominally significant RD-associated genes, excluding the synonymous model. These genes were labelled "any non-synonymous" genes. Significant genes identified with the synonymous model were utilized as the negative control group. Known ocular disease risk genes were separated into AR and AD inheritance patterns based on the Genomics England "Level 2-4" columns, which document phenotypes with their documented inheritance patterns. "Biallelic" was labelled as AR and "monoallelic" was labelled as AD. Genes were allowed to have both AR and AD labels. Overlapping gene lists from the nominally significant RD-associated genes, AD and AR ocular disease risk genes were visualized in a venn diagram using the DeepVenn web tool. Fisher's exact test with FDR correction for multiple comparisons was used for overrepresentation analysis of nominally associated RD-associated genes and ocular disease risk genes. The resulting odds ratios and 95% confidence intervals were used to generate a forest plot.

## Supporting information

**S1 Fig. QQ plots for the variant-level ExWAS. (A-C)** Observed versus expected $-\log_{10}(p)$ values for the additive, dominant, and recessive genetic models. The null-distribution of expected p-values is based on an n-of-1 permutation of case-control labels.
(TIF)

**S2 Fig. QQ plots for the gene-level collapsing analysis.** Observed versus expected $-\log_{10}(p)$ values for all 11 collapsing analysis models. The null-distribution of expected p-values is based on an n-of-1 permutation of case-control labels.
(TIF)

**S3 Fig. Functional assessment of *VSX2* variants in APRE-19 cells. A)** Representative fluorescent images of APRE-19 cells transduced with lentiviruses expressing WT or variant *VSX2* constructs or a control vector (Blue = DAPI, green = eGFP). **B)** Quantification of cell area (µm 2) across experimental conditions. **C)** Principal component analysis of bulk RNA-seq data demonstrating distinct transcriptional clustering of APRE-19 cells expressing different *VSX2* variants. **D)** Differential expression analysis comparing WT vs control, R227W vs WT, and E218D vs WT. **E)** Differential expression of *MITF* and **F)** WNT signaling genes across experimental conditions.
(TIF)

**S4 Fig. *VSX2* expression during human embryonic development. A)** UMAP of developing human retinal cell types and **B)** UMAP highlighting *VSX2* expression.
(TIF)

**S5 Fig. Density plot of age for RD cases and controls.** Note: this density plot was generated before age filtering and down-sampling in the All of Us Biobank.
(TIF)

**S1 Table. Variant-level associations from the Euroepan ExWAS that achieved p<1x10<sup>-4</sup>.** Abbreviations: oddsLCI and oddsUCI indicate the lower and upper bounds of the 95% confidence interval for the odds ratio, respectively.
(XLSX)

**S2 Table. Collapsing analysis qualifying variant models.**
(XLSX)

**S3 Table. Gene-level associations from the European collapsing analysis that achieved p<1x10<sup>-4</sup>.**
(XLSX)

**S4 Table. Gene-level associations from the pan-ancestry collapsing analysis that achieved p<1x10<sup>-4</sup>.**
(XLSX)

**S5 Table. Replication ExWAS and collapsing analyses for RD cases vs controls with respect to *VSX2* carrier status in the All of Us and Genomics England Biobanks.**
(XLSX)

**S6 Table. Incidence of co-occurring ICD10 codes in UKBiobank case-controls.**
(XLSX)

**S7 Table. Adjusting ExWAS and collapsing analysis results with mean spherical equivalent (MSE).**
(XLSX)

**S8 Table. Adjusting ExWAS and collapsing analysis results after removing individuals with cataract diagnosis prior to RD or individuals with glaucoma diagnosis.**
(XLSX)

**S9 Table. Nominal gene-level associations between *VSX2* and OCT measurements under the flexnonsynmtr collapsing model.**
(XLSX)

**S10 Table. Nominal variant-level associations between the *VSX2* p.Glu218Asp variant and OCT measurements under the dominant ExWAS model.**
(XLSX)

**S11 Table. All previously reported disease-associated variants in *VSX2*.** Abbreviations: AA (amino acid).
(XLSX)

**S12 Table. *VSX2* flexnonsynmtr qualifying variants observed in RD cases.**
(XLSX)

**S13 Table. Differential gene expression analysis for comparing WT VSX2 vs control lentivirus APRE-19 cells.**
(XLSX)

**S14 Table. Differential gene expression analysis for comparing R227W vs control.**
(XLSX)

**S15 Table. Differential gene expression analysis for comparing E218D vs control.**
(XLSX)

**S16 Table. Differential gene expression analysis for comparing R227W vs WT VSX2.**
(XLSX)

**S17 Table. Differential gene expression analysis for comparing E218D vs WT.**
(XLSX)

**S18 Table. Differential gene expression analysis for comparing E218D vs R228W.**
(XLSX)

**S19 Table. Nominally significant genes from the collapsing analysis (p < 0.05) that overlap with known Mendelian eye disease genes.** Inheritance patterns are recorded in the "AR_genes" and "AD_genes" columns.
(XLSX)

## Author contributions

**Conceptualization:** Daniel C. Brock, Justin S. Dhindsa, Likhita Nandigam, Slavé Petrovski, Ryan S. Dhindsa.

**Data curation:** Jonathan Mitchell, Fengyuan Hu, Xiaoyin Li, Quanli Wang, Ryan S. Dhindsa.

**Formal analysis:** Daniel C. Brock, Justin S. Dhindsa, Vida Ravanmehr, Jonathan Mitchell, Fengyuan Hu, Xiaoyin Li, Quanli Wang, Kevin Wu, Jessica C. Butts, Ryan S. Dhindsa.

**Investigation:** Daniel C. Brock, Justin S. Dhindsa, Yifan Chen, Vida Ravanmehr, Jonathan Mitchell, Fengyuan Hu, Xiaoyin Li, Quanli Wang, Ryan S. Dhindsa.

**Project administration:** Slavé Petrovski, Ryan S. Dhindsa.

**Resources:** Jessica C. Butts, Slavé Petrovski, Ryan S. Dhindsa.

**Software:** Jonathan Mitchell, Fengyuan Hu, Xiaoyin Li, Quanli Wang.

**Supervision:** Slavé Petrovski, Ryan S. Dhindsa.

**Validation:** Vida Ravanmehr.

**Visualization:** Daniel C. Brock, Justin S. Dhindsa, Ryan S. Dhindsa.

**Writing – original draft:** Daniel C. Brock, Justin S. Dhindsa, Likhita Nandigam, Ryan S. Dhindsa.

**Writing – review & editing:** Daniel C. Brock, Justin S. Dhindsa, Hardeep S. Dhindsa, Benjamin J. Frankfort, Nicholas M. Tran, Slavé Petrovski, Ryan S. Dhindsa.

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
