## [Decision Letter · Decision Letter 0]

17 Sep 2025

PGENETICS-D-25-00816

Rare heterozygous missense variants in VSX2 are associated with retinal detachment

PLOS Genetics

Dear Dr. Dhindsa,

Thank you for submitting your manuscript to PLOS Genetics. After careful consideration, we feel that it has merit but does not fully meet PLOS Genetics's publication criteria as it currently stands. Therefore, we invite you to submit a revised version of the manuscript that addresses the points raised during the review process.

Please submit your revised manuscript within 60 days Nov 16 2025 11:59PM. If you will need more time than this to complete your revisions, please reply to this message or contact the journal office at plosgenetics@plos.org. Please include the following items when submitting your revised manuscript:

We look forward to receiving your revised manuscript.

Kind regards,

Stuart A Scott, PhD

Academic Editor

PLOS Genetics

Hua Tang

Section Editor

PLOS Genetics

Aimée Dudley

Editor-in-Chief

PLOS Genetics

Anne Goriely

Editor-in-Chief

PLOS Genetics

**Journal Requirements:**

At this stage, the following Authors/Authors require contributions: Daniel C. Brock, Justin S. Dhindsa, Vida Ravanmehr, Jonathan Mitchell, Fengyuan Hu, Xiaoyin Li, Likhita Nandigam, Quanli Wang, Hardeep S. Dhindsa, Benjamin J. Frankfort, Nicholas M. Tran, Slavé Petrovski, and Ryan Dhindsa. Please ensure that the full contributions of each author are acknowledged in the "Add/Edit/Remove Authors" section of our submission form.

The list of CRediT author contributions may be found here: https://journals.plos.org/plosgenetics/s/authorship#loc-author-contributions

https://journals.plos.org/plosgenetics/s/submission-guidelines#loc-parts-of-a-submission

4) Thank you for including an Ethics Statement for your study. Please state whether the consent obtained is verbal or written.

5) Please upload all main figures as separate Figure files in .tif or .eps format. For more information about how to convert and format your figure files please see our guidelines:

6) We have noticed that you have uploaded Supporting Information files, but you have not included a list of legends in the manuscript. Please add a full list of legends for your Supporting Information files after the references list.

7) Your current Financial Disclosure states, "The author(s) received no specific funding for this work."

However, your funding information on the submission form indicates receiving a fund. Please ensure that the funders and grant numbers match between the Financial Disclosure field and the Funding Information tab in your submission form. Note that the funders must be provided in the same order in both places as well.

Please amend your detailed Financial Disclosure statement. This is published with the article. It must therefore be completed in full sentences and contain the exact wording you wish to be published.

1) Please clarify all sources of financial support for your study. List the grants, grant numbers, and organizations that funded your study, including funding received from your institution. Please note that suppliers of material support, including research materials, should be recognized in the Acknowledgements section rather than in the Financial Disclosure

2) State the initials, alongside each funding source, of each author to receive each grant. For example: "This work was supported by the National Institutes of Health (####### to AM; ###### to CJ) and the National Science Foundation (###### to AM)."

3) State what role the funders took in the study. If the funders had no role in your study, please state: "The funders had no role in study design, data collection and analysis, decision to publish, or preparation of the manuscript."

4) If any authors received a salary from any of your funders, please state which authors and which funders.

8) Thank you for stating "J.M., F.H., X.L., Q.W, and S.P. are current employees and/or shareholders of

AstraZeneca. R.S.D. has received consulting fees from AstraZeneca." Please declare all competing interests beginning with the statement "I have read the journal's policy and the authors of this manuscript have the following competing interests:"

**Reviewers' comments:**

Reviewer's Responses to Questions

Reviewer #1: This article discusses the results of a large-scale association study using data from the UK biobank regarding the potential role of VSX2 in retinal detachment. Overall, the study is well-designed, and the article is well written. I suggest a few minor edits.

1- A more detailed description of the genetic architecture of RD is warranted. While the authors quickly mention that in the intro, I believe a more detailed overview of what has been known so far would educate the readers better.

2- Line 84: A reference is needed for evidence for retinal dystrophies and cataract (role of het variants)

3- What other ICD10 codes related to eye disease were found in the RD cohort? What measures were taken to make sure no confounding phenotypes were present in those patients (did any have cataracts, glaucoma, retinal degeneration, etc.).

4- Some abbreviations have not been described e.g. RPRS. Please make sure all abbreviations are detailed the first time they are mentioned.

Reviewer #2: Retinal detachment (hereafter “RD”)is a potentially blinding condition that arises as a result of the a widening gap between the neuroretina and the retinal pigmented epithelium. Prompt intervention may stabilize the condition, but not reverse any of its consequences that should medical attention be delayed, could be severe.

Statistical association between disease and genes are very useful not only because at a certain stage they may enable the prediction of individuals who are at higher disease risk and on whom medical attention can be prioritized, but they also inform about the underlying pathophysiological mechanisms underlying that condition, creating the premises for newer interventions and therapies.

While most genetic associations have focused on common variants (the so called “common variant – common disease” model), this approach tends to omits some of the strongest clinical effect is generally inversely correlated with allele frequency, because rare variant characterization is limited by cost and technological availability. Increasingly population-based cohorts are offering opportunities for rare variant analyses, and these opportunities have been picked up by Brock and co-authors, who analyzed available exome sequencing information from the UK Biobank and two smaller (but in absolute terms large) cohorts.

The authors find association between rare coding variants and RD, and in addition to known genes implicated in RD, such as FAT3, RDH5 and COL2A1they for the first time report an association with a missense variant within the coding sequence VSX2. Evidence that other missense variants within that gene may contribute at various degrees to RD is also reported. The authors conclude that these associations are proper to RD and not confounded from associations with other phenotypes.

This is a detailed work where the reader will find so much to admire. There are however question marks and potential limitations on which this review will be concentrating, in the hope of improving the clarity and potentially the impact of the report.

The main question that remains after reading the manuscript is whether the VSX2 Glu218Asp mutation is a primary RD gene or does so through its putative action through other channels. Results by Boutin et al. have found a very strong genetic correlation between RD on one side and both refractive error (r=0.25) and a cataract (r=0.26) on the other. This means that simple genetic risk scores calculated for any these conditions would be associated with RD – but whether they inform about RD pathogenesis may be more debatable. Variants within VSX2 (previously known as CHX10) have been previously linked to a syndromic combination of microphthalmia, cataract and iris colobomas. The variant identified in the originally reported families codes for the same protein domain, just 18 aminoacids away. It is possible therefore that VSX2 may cause RD indirectly, through its effects on refractive error AND cataract. The authors have explored one of these routes (myopia) but are not reporting any association with history of cataract operations. Stratifying for history of cataract operation would have been of great interest for the reader.

Another question pertains to certain statistical analyses choices made by the authors. Controlling for potential confounders is very important, but this is done in ways which may be unusual and occasionally inconsistent. For example, the authors correctly used Mantel-Hanetzel stratified analyses, but they sex-match cases and controls, without analyzing them separately. They do not control for age, a well known risk factor, and rather selectively don’t report the age distribution among controls. It is unclear why slightly more tortuous solutions were found for situations in which many statisticians would have either stratified analyses, or linear / mixed model analyses, also adjusting for age.

Improving clarity on this issue is important for at least two reasons. First, the risk of ascertainment bias in cases and controls, and second, the absence of any description of the matching procedures could lead some readers to doubt whether there is any implicit risk of selection bias towards particular age-groups.

The authors report the results of an analysis of “overrepresentation” and “enrichment” of variants linked to other ocular conditions with RD. Some readers may not agree with the application of these labels to the analyses conducted. An argument can be made that these are simple pleiotropy tests and a formal statistical test for enrichment or more-than-expected occurrence of these statistical associations is not reported. It is reasonable to expect some overrepresentation due to the multifactorial nature of RD etiology, but evidence presented is not unambiguously clear.

Less important instances where more clarity would be beneficial are below. The authors report checking concordance with previous genotypes released for the same individuals. It may be better to explain the logic behind this, but also the choice of the thresholds: 0.49 and 0.465, it is difficult to understand why such choices were made. Again, an argument could have been made to remove all samples that were not 99% concordant with previous genotyping versions of themselves, but it is possible there is a problem of clarity and the way the thoughts were phrased.

The authors write that “there were significant (p<1x10-8) associations” and that the genomic inflation factors were reasonably tame. Both these notions are linked in different ways to expectations about linkage disequilibrium in a population. While there is clarity of what the level of significance we should expect for common variants, it would be useful to have a description of how the authors picked that level of significance for their study.

The authors write that “a stronger association with serous RD (p=5.6x10-5; OR [95% CI] = 3.4 [1.9-5.8]) than with rhegmatogenous RD (p=2.7x10-4; OR [95% CI] = 2.6 [1.5-4.1])”. Yet the confidence intervals largely overlap.

Tables and figures legends may be improved. For example "FF, FT and TT, OddsUCI" in one of the tables are not accronyms the reader would immediately relate to.

Reviewer #3: Brock and Dhindsa et al. details a genome-wide sequencing study on retinal detachment, using 7,000 cases from the UK Biobank and replication cohorts. The authors show rare heterozygous missense variants in VSX2 as a risk factor for RD, a gene previously associated with recessive developmental eye disorders.

Major Comments:

Rare variants typically confer larger effect sizes and can provide clear mechanism and

clinical impacts than on common variants. It would be helpful to review the following manuscript in light of this work:

• Lupski, J. R., Belmont, J. W., Boerwinkle, E. & Gibbs, R. A. Clan genomics and the complex architecture of human disease. Cell 147, 32–43 (2011).

The gene dosage model for VSX2 is well documented and discussed in the manuscript. It would be helpful to more explicitly connect these findings other heterozygous variants in recessive disease genes manifesting as milder, or late-onset phenotypes. Possibly detail these other examples in the manuscript in the discussion.

Were copy number variants or other structural variants looked for or studied? This may be helpful to find other cases with VSX2 variation.

It is interesting that the effect size estimates (particularly for the p.Glu218Asp variant) differ so much between the different cohorts. It would be useful to discuss the potential reasons for this difference (sequencing depth, phenotypic delineation differences, etc?)

Additionally, have you thought about any functional validation especially for the p.Glu218Asp variant?

Have you looked through ClinVar or other clinical variant databases to see how VSX2 is represented?

• Landrum, M. J. et al. ClinVar: improving access to variant interpretations and supporting evidence. Nucleic Acids Res. 46, D1062–D1067 (2018).

**Have all data underlying the figures and results presented in the manuscript been provided?**

Reviewer #1: Yes

Reviewer #2: Yes

Reviewer #3: Yes

PLOS authors have the option to publish the peer review history of their article (what does this mean? ). If published, this will include your full peer review and any attached files.

**Do you want your identity to be public for this peer review?** For information about this choice, including consent withdrawal, please see our Privacy Policy .

Reviewer #1: No

Reviewer #2: No

Reviewer #3: No

**Figure resubmission:**

**Reproducibility:**



---

## [Decision Letter · Decision Letter 1]

9 Jan 2026

Dear Dr Dhindsa,

We are pleased to inform you that your manuscript entitled "Rare heterozygous missense variants in *VSX2*  are associated with retinal detachment" has been editorially accepted for publication in PLOS Genetics. Congratulations!

Yours sincerely,

Stuart A Scott, PhD

Academic Editor

PLOS Genetics

Hua Tang

Section Editor

PLOS Genetics

Aimée Dudley

Editor-in-Chief

PLOS Genetics

Anne Goriely

Editor-in-Chief

PLOS Genetics

BlueSky: @plos.bsky.social

Comments from the reviewers (if applicable):

Reviewer's Responses to Questions

**Comments to the Authors:**

Reviewer #1: Thank you for addressing the reviewers' questions and incorporating suggested changes. I believe such changes improved the manuscript quality and should make the readers experience better.

Reviewer #3: The authors have addressed all of my previous concerns in this revised version.

**Have all data underlying the figures and results presented in the manuscript been provided?**

Reviewer #1: Yes

Reviewer #3: Yes

PLOS authors have the option to publish the peer review history of their article (what does this mean? ). If published, this will include your full peer review and any attached files.

**Do you want your identity to be public for this peer review?** For information about this choice, including consent withdrawal, please see our Privacy Policy .

Reviewer #1: **Yes:** Sherin Shaaban

Reviewer #3: No

**Data Deposition**

http://datadryad.org/submit?journalID=pgenetics&manu=PGENETICS-D-25-00816R1

**Press Queries**

---

## [Editor Report · Acceptance letter]

PGENETICS-D-25-00816R1

Rare heterozygous missense variants in *VSX2*  are associated with retinal detachment

Dear Dr Dhindsa,

We are pleased to inform you that your manuscript entitled "Rare heterozygous missense variants in *VSX2*  are associated with retinal detachment" has been formally accepted for publication in PLOS Genetics! Your manuscript is now with our production department and you will be notified of the publication date in due course.

With kind regards,

Judit Kozma

PLOS Genetics

On behalf of:
